# TransNetR: Transformer-based Residual Network for Polyp Segmentation with Multi-Center Out-of-Distribution Testing

**Debesh Jha**[1]                                                    DEBESH.JHA@NORTHWESTERN.EDU
[1] *Department of Radiology, Northwestern university*
**Nikhil Kumar Tomar**[1]                                    NIKHIL.TOMAR@NORTHWESTERN.EDU
**Vanshali Sharma**[2]                                              VANSHALISHARMA@IITG.AC.IN
[2] *Indian Institute of Technology Guwahati*
**Ulas Bagci**[1]                                                        ULAS.BAGCI@NORTHWESTERN.EDU

**Editors:** Accepted for publication at MIDL 2023

## Abstract

Colonoscopy is considered the most effective screening test to detect colorectal cancer (CRC) and its precursor lesions, i.e., polyps. However, the procedure experiences high miss rates due to polyp heterogeneity and inter-observer dependency. Hence, several deep learning powered systems have been proposed considering the criticality of polyp detection and segmentation in clinical practices. Despite achieving improved outcomes, the existing automated approaches are inefficient in attaining real-time processing speed. Moreover, they suffer from a significant performance drop when evaluated on inter-patient data, especially those collected from different centers. Therefore, we intend to develop a novel real-time deep learning based architecture, Transformer based Residual network (TransNetR), for colon polyp segmentation and evaluate its diagnostic performance. The proposed architecture, TransNetR, is an encoder-decoder network that consists of a pre-trained ResNet50 as the encoder, three decoder blocks, and an upsampling layer at the end of the network. TransNetR obtains a high dice coefficient of 0.8706 and a mean Intersection over union of 0.8016 and retains a real-time processing speed of 54.60 on the Kvasir-SEG dataset. Apart from this, the major contribution of the work lies in exploring the generalizability of the TransNetR by testing the proposed algorithm on the out-of-distribution (test distribution is unknown and different from training distribution) dataset. As a use case, we tested our proposed algorithm on the PolypGen (6 unique centers) dataset and two other popular polyp segmentation benchmarking datasets. We obtained state-of-the-art performance on all three datasets during out-of-distribution testing. The source code of TransNetR is publicly available at `https://github.com/DebeshJha/TransNetR`.

**Keywords:** Out-of-distribution generalization, Out-of-distribution testing, Transformer, Polyp segmentation, Residual network, PolypGen

## 1. Introduction

Colorectal cancer (CRC) is the third most prevalent malignancy and accounts for 9.4% of cancer-related deaths worldwide (Sung et al., 2021). The alarmingly increasing cases of CRC have led to the adoption of various screening tests (Kanth and Inadomi, 2021) to lower the risk of its incidence and related mortalities. The colonoscopy procedure is the most preferred among these tests, which allows clinicians to identify and examine CRC precursor lesions, i.e., polyps. The early detection and resection of such polyps are crucial to

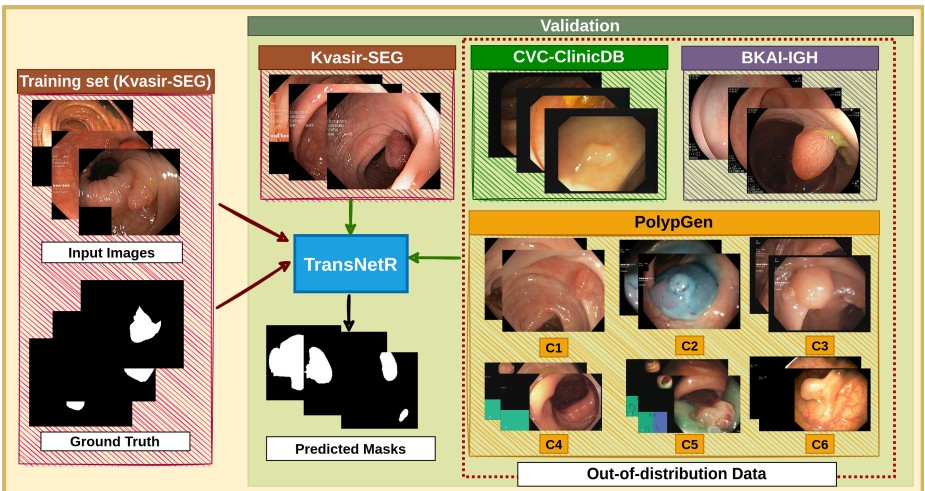

Figure 1: Illustration of different scenarios expected to arise in real-world settings. The proposed work conducted both in-distribution and out-of-distribution validation process. C1 to C6 represent the different centers data present in PolypGen (Ali et al., 2023) dataset.

prevent them from developing into cancer at their later stages. An efficient removal process of polyps requires that the clinicians have access to their accurate location information and precise boundary details. Thus, in clinical settings, polyp segmentation is a crucial task.

Despite the wide acceptance of colonoscopy as the gold standard for CRC screening, the associated traditional assessment procedures experience significant polyp miss rates attributed to various factors (Kim et al., 2017). First, the process involves dependency on the operator's experience and the risk of overlooking polyps due to faster colonoscope withdrawal time. Second, high variations in polyps' appearance, such as color, size, and shape, further complicate the detection task. Third, the lack of intense contrast between fuzzy polyp boundaries and surrounding mucosa makes polyps camouflaged (Fan et al., 2021) against other endoluminal structures. These challenging factors present a need for automated systems to perform polyp segmentation, which could complement gastroenterologists' ability to detect and delineate polyp boundaries for accurate resection.

In this context, many deep learning (DL) based techniques (Akbari et al., 2018; Duc et al., 2022) have been developed in the past few years. Although these methods have reported considerable improvements over manual assessments and hand-crafted features based approaches (Bernal et al., 2015; Tajbakhsh et al., 2015), they still lack generalizability. As a result, the automated techniques proposed so far are considerably prone to limited performance under real-world scenarios where they are expected to be utilized for different patients, hospitals with varying imaging modalities, or even across varied populations. Such cases involve significant variations in colon polyps due to demographic changes, including gender, age, race, and region (Yang et al., 2020). For example, a study (Cekodhima et al., 2016) observed polyp morphology and its malignant potential to be related to a patient's age.

Similar disparities are reported based on geographical distribution, where the prevalence and location of large polyps are affected by race and ethnicity (Lieberman et al., 2014).

Apart from population variations, the differences in colonoscopy conducting centers and their associated video-capturing modality types also create domain shift problems (Chen et al., 2021). Some sample images collected from different datasets and multiple centers are shown in Figure 1. The figure illustrates different validation scenarios and also demonstrates the range of heterogeneity possessed by polyps. Although the segmentation performance achieved by SOTA methods is noteworthy, the issue of generalizability remains marginally explored as compared to testing the performance on in-distribution (iD) polyp data. One of the reasons might be the lack of availability of multi-center datasets. In this work, we propose a DL model, which is a transformer based residual network (TransNetR), to achieve accurate and real-time polyp segmentation and to generalize well on unseen out-of-distribution (OOD) data. Our architecture is inspired by the remarkable success of encoder-decoder structures, Transformers and residual learning in biomedical image analysis. To validate the efficacy of our algorithm, we test the algorithm on different datasets, which are collected from various parts of the world and unique from our training samples. The main contributions of our work are summarized below:

- We present a novel DL based polyp segmentation model. The architecture integrates the strength of transformer and residual learning to generate precise segmentation outcomes even while testing on OOD data and maintains high performance along with real-time processing speed, which is important for clinical intergration.

- The proposed architecture is extensively validated on iD and OOD datasets. The obtained results from three datasets (8 unique centers) signifies that the model performs consistently well on datasets from unseen clinical centers, showing a better generalization ability as compared to the other SOTA approach.

## 2. Method

In this section, we will present the proposed architecture in detail along with its components.

### 2.1. TransNetR

Figure 2 shows the block diagram of the proposed TransNetR. As observed in the figure, TransNetR is an encoder decoder network which begins with a pre-trained ResNet50 as the encoder. We pass input image to the pre-trained encoder and extract four different intermediate feature maps from it. These intermediate feature maps are then passed through a $1 \times 1$ convolution layer, which is followed by a batch normalization and a LeakyReLU activation function. The $1 \times 1$ convolution layer helps in reducing the number of output feature channels which reduces the number of parameters. Next follows the decoder network, which contains three decoder blocks. The reduced feature map is fed to the first decoder block, where it is first passed through a bilinear upsampling layer. The upsampling layer increases the spatial dimensions of the feature maps by a factor of two. The upsampled feature map is then concatenated with the next reduced feature map having the same spatial dimensions.

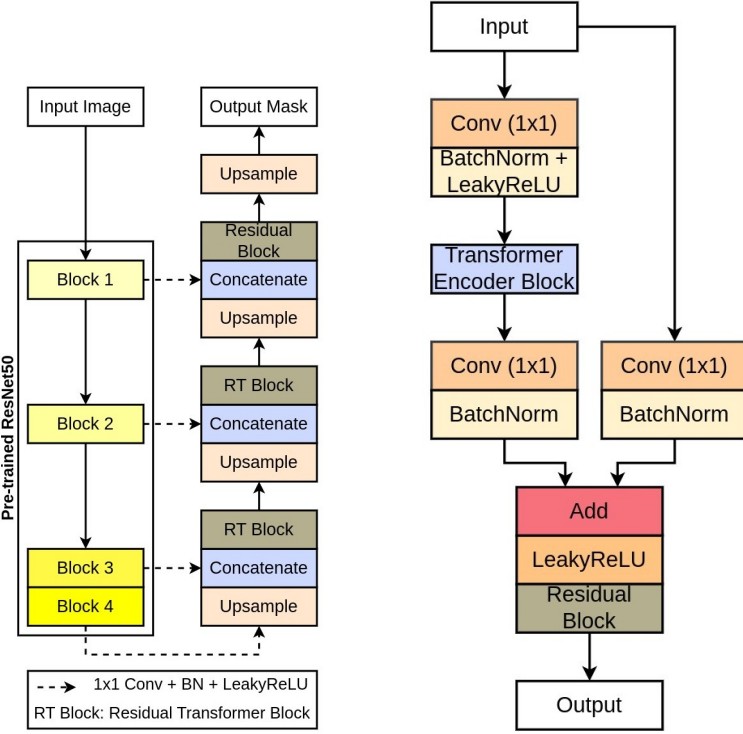

Figure 2: Block diagram of TransNetR along with the Residual Transformer (RT) block.

The concatenation creates a shortcut connection from the pre-trained encoder to the decoder block, which helps in a better flow of information from encoder to decoder. The short connection fetches the feature maps, which might be lost due to the depth of the network. The concatenated feature maps are further passed through our proposed Residual Transformer block, where the feature maps are first reshaped into patches and then passed to the transformer layers. These layers consist of the multi-head self-attention, which helps in learning better feature representation. Subsequently, the output from the first decoder block is passed to the second and then to the final decoder block. In the final decoder block, the Residual Transformer block is replaced with a simple residual block to reduce the number of trainable parameters. The output from the final decoder is passed through a bilinear upsampling layer which increases the spatial dimensions of the feature maps by a factor of two. The upsampled feature map is then passed through a $1 \times 1$ convolution layer with a sigmoid activation function.

## 2.2. Residual Transformer Block

The Residual Transformer block begins with a $1 \times 1$ convolution layer, followed by a batch normalization and a LeakyReLU activation function. After that, we flattened the feature maps, where we use a constant patch size of four. The flattened feature maps are then passed to the transformer block, having four heads and two layers. The transformer block provides self-attention on the feature maps, which makes the network more robust. The output from

Table 1: Quantitative results on the Kvasir-SEG test dataset. The parameters are in Millions and Flops are in GMac.

| Method | mIoU | DSC | Rec. | Prec. | F2 | FPS | Para. | Flops |
|---|---|---|---|---|---|---|---|---|
| U-Net (Ronneberger et al., 2015) | 0.7472 | 0.8264 | 0.8504 | 0.8703 | 0.8353 | **106.88** | 31.04 | 54.75 |
| U-Net++ (Zhou et al., 2018) | 0.7420 | 0.8228 | 0.8437 | 0.8607 | 0.8295 | 81.34 | 9.16 | 34.65 |
| ResU-Net++ (Jha et al., 2019) | 0.5341 | 0.6453 | 0.6964 | 0.7080 | 0.6576 | 43.11 | 4.06 | 15.81 |
| HarDNet-MSEG (Huang et al., 2021) | 0.7459 | 0.8260 | 0.8485 | 0.8652 | 0.8358 | 34.80 | 33.34 | 6.02 |
| ColonSegNet (Jha et al., 2021) | 0.6980 | 0.7920 | 0.8193 | 0.8432 | 0.7999 | 73.95 | 5.01 | 62.16 |
| UACANet (Kim et al., 2021) | 0.7692 | 0.8502 | 0.8799 | 0.8706 | 0.8626 | 25.85 | 69.16 | 31.51 |
| UNeXt (Valanarasu and Patel, 2022) | 0.6284 | 0.7318 | 0.7840 | 0.7656 | 0.7507 | 87.47 | **1.47** | **0.57** |
| **TransNetR (Ours)** | **0.8016** | **0.8706** | **0.8843** | **0.9073** | **0.8744** | 54.60 | 27.27 | 10.58 |

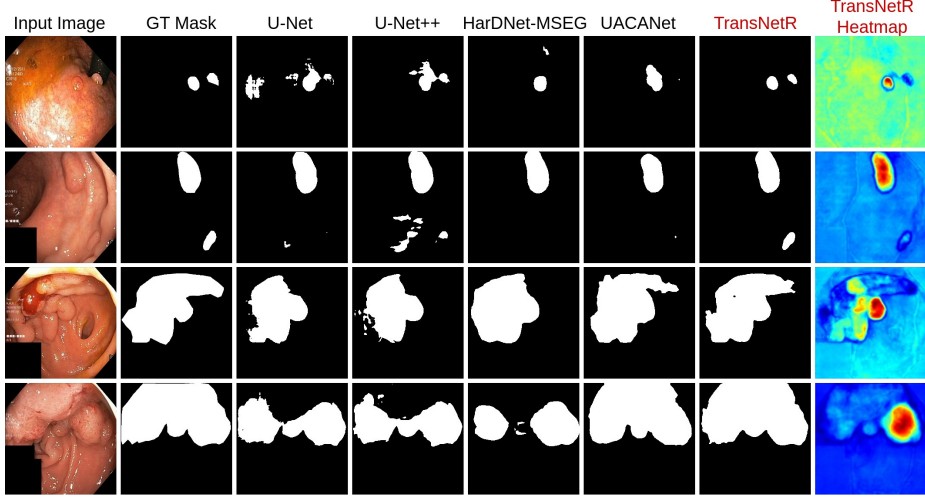

Figure 3: Qualitative example showing polyp segmentation on Kvasir-SEG (Jha et al., 2020).

the transformer block is then reshaped back in the same shape as the input. Next, the feature map is passed through a $1 \times 1$ convolution, followed by batch normalization. After that, it is followed by addition with the input feature maps and then passed through the LeakyReLU activation function. Finally, the output from the LeakyReLU is passed through a residual block which acts as the output of the Residual Transformer block.

## 3. Experiments and Results

### 3.1. Dataset details and Experiment setup

We evaluate our model's performance using four datasets; namely, Kvasir-SEG (Jha et al., 2020), PolypGen (Ali et al., 2023), CVC-ClinicDB (Bernal et al., 2015), and BKAI-IGH (Ngoc Lan et al., 2021). Kvasir-SEG consisting of 1000 images, is used for training purposes. We use 880 images in the training split, and the rest are reserved for testing. We

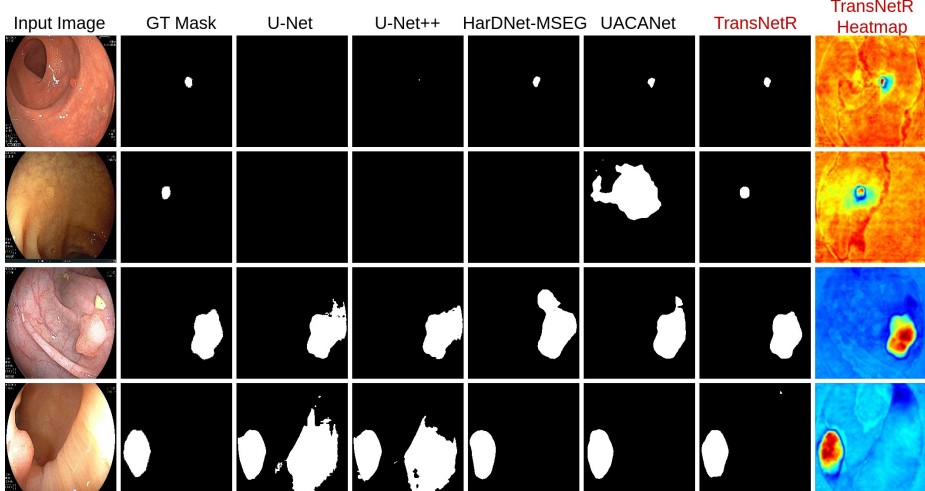

Figure 4: Cross-data result when models trained on Kvasir-SEG & tested on BKAI-IGH.

Table 2: Results of the models when trained on Kvasir-SEG and tested on OOD dataset.

| Method | mIoU | mDSC | Recall | Precision | F2 |
|---|---|---|---|---|---|
| **Training dataset: Kvasir-SEG − Test dataset: CVC-ClinicDB** | | | | | |
| U-Net (Ronneberger et al., 2015) | 0.5433 | 0.6336 | 0.6982 | 0.7891 | 0.6563 |
| U-Net++ (Zhou et al., 2018) | 0.5475 | 0.6350 | 0.6933 | 0.7967 | 0.6556 |
| ResU-Net++ (Jha et al., 2019) | 0.3585 | 0.4642 | 0.5880 | 0.5770 | 0.5084 |
| HarDNet-MSEG (Huang et al., 2021) | 0.6058 | 0.6960 | 0.7173 | 0.8528 | 0.7010 |
| ColonSegNet (Jha et al., 2021) | 0.5090 | 0.6126 | 0.6564 | 0.7521 | 0.6246 |
| UACANet (Kim et al., 2021) | 0.6808 | **0.7659** | **0.7639** | 0.8820 | **0.7599** |
| UNeXt (Valanarasu and Patel, 2022) | 0.3901 | 0.4915 | 0.6125 | 0.6609 | 0.5318 |
| **TransNetR (Ours)** | **0.6912** | 0.7655 | 0.7571 | **0.9200** | 0.7565 |
| **Training dataset: Kvasir-SEG − Test dataset: BKAI-IGH** | | | | | |
| U-Net (Ronneberger et al., 2015) | 0.5686 | 0.6347 | 0.6986 | 0.7882 | 0.6591 |
| U-Net++ (Zhou et al., 2018) | 0.5592 | 0.6269 | 0.6900 | 0.7968 | 0.6493 |
| ResU-Net++ (Jha et al., 2019) | 0.3204 | 0.4166 | 0.6979 | 0.3922 | 0.5019 |
| HarDNet-MSEG (Huang et al., 2021) | 0.5711 | 0.6502 | **0.7420** | 0.7469 | **0.6830** |
| ColonSegNet (Jha et al., 2021) | 0.4910 | 0.5765 | 0.7191 | 0.6644 | 0.6225 |
| UACANet (Kim et al., 2021) | 0.5734 | 0.6531 | 0.7361 | 0.7689 | 0.6790 |
| UNeXt (Valanarasu and Patel, 2022) | 0.3304 | 0.4156 | 0.6085 | 0.4933 | 0.4722 |
| **TransNetR (Ours)** | **0.5998** | **0.6601** | 0.6660 | **0.9072** | 0.6584 |

performed extensive data augmentations to obtain more training samples. The cross-dataset performance is validated by evaluating the model on the other three datasets, where CVC-ClinicDB, BKAI-IGH, and PolypGen contain 612, 1000, and 1537 still images, respectively. It is worth mentioning that PolypGen incorporates data collected from six different centers covering varied populations. Thus, validation of the proposed algorithm on these types of OOD datasets makes the study more comprehensive and closer to a real-world scenario.

The proposed model is implemented using the Pytorch framework, and experiments are conducted on an NVIDIA RTX 3090 GPU system. An adam optimizer with a learning rate of $1e^{-4}$ is used, and batch size is set to 8. The loss function used is a combination of binary

Table 3: Results of models trained on Kvasir-SEG & tested on **PolypGen 23 videos**.

| Method | mIoU | mDSC | Recall | Precision | F2 |
|---|---|---|---|---|---|
| **Training dataset: Kvasir-SEG – Test dataset: PolypGen Video Sequence** | | | | | |
| U-Net (Ronneberger et al., 2015) | 0.4049 | 0.4559 | 0.6307 | 0.5762 | 0.4668 |
| U-Net++ (Zhou et al., 2018) | 0.4272 | 0.4772 | 0.6198 | 0.6269 | 0.4876 |
| ResU-Net++ (Jha et al., 2019) | 0.1589 | 0.2105 | 0.5095 | 0.2447 | 0.2303 |
| HarDNet-MSEG (Huang et al., 2021) | 0.4171 | 0.4662 | 0.6217 | 0.6120 | 0.4757 |
| ColonSegNet (Jha et al., 2021) | 0.3058 | 0.3574 | 0.5296 | 0.4804 | 0.3533 |
| UACANet (Kim et al., 2021) | 0.4155 | 0.4748 | **0.6357** | 0.6108 | 0.4886 |
| UNeXt (Valanarasu and Patel, 2022) | 0.2457 | 0.2998 | 0.5658 | 0.3661 | 0.3201 |
| **TransNetR (Ours)** | **0.4717** | **0.5168** | 0.5777 | **0.7881** | **0.5105** |

Table 4: Ablation study of the proposed TransNetR on the Kvasir-SEG dataset.

| Method | mIoU | mDSC | Recall | Precision |
|---|---|---|---|---|
| TransNetR without RT block | 0.7882 | 0.8629 | 0.8841 | 0.8923 |
| TransNetR (RT block replaced with residual block) | 0.7977 | 0.8669 | 0.8833 | 0.8953 |
| **TransNetR (Ours)** | **0.8016** | **0.8706** | **0.8843** | **0.9073** |

cross-entropy and dice loss. We quantitatively compared the performance of TransNetR with SOTA methods using widely used evaluation metrics, such as mIoU, mDSC, Recall, Precision, F2, and processing speed (FPS).

## 3.2. Performance Evaluation

We have evaluated TransNetR performance in different scenarios. Firstly, we conducted validation tests to investigate the model's learning ability with seen data, i.e., the test split of Kvasir-SEG. This is followed by OOD testing for the generalizability test.

**Learning ability:** The results associated with the seen dataset are presented in Table 1. Our proposed method reported the best outcome relative to other approaches with mIoU of 0.8016, mDSC of 0.8706, recall of 0.8843, precision of 0.9073 and F2 score of 0.8744. The performance of UACANet (Kim et al., 2021) is competitive with our method. However, TransNetR outperforms UACANet by 3.24% in mIoU and 2.04% in mDSC. Moreover, the inference time of our model is 54.60 FPS which is twice of UACANet. The qualitative results are shown in Figure 3. It can be observed that our model has correctly segmented the polyp regions even when there is multiple polyps in the image frame and has captured relatively more accurate boundary details as compared to UACANet.

**Generalization ability:** We investigated the OOD outcomes that are presented in Table 2, Table 3, Table 5, and Appendix Table 6. The consistent superior performance of our model on all three still frame datasets and 23 video sequence from PolypGen confirms its better generalization ability on different data distributions. Besides achieving a substantial difference on PolypGen (see Table 5), our proposed model outperformed the next closest competitive model, UACANet (Kim et al., 2021), by a significant margin on other polyp benchmarking datasets as well. Quantitatively, an improvement of 2.81% in PolypGen, 5.62% in PolypGen Video sequence, 1.04% in CVC-ClinicDB, and 2.64% in BKAI-IGH are reported when evaluated in terms of mIoU. Additionally, we have presented center-wise results because *PolypGen images come from different centers, and each center is independent.*

Table 5: Results of the models when trained on Kvasir-SEG and tested on OOD dataset.

| Method | mIoU | mDSC | Recall | Precision | F2 |
|---|---|---|---|---|---|
| **Training dataset: Kvasir-SEG − Test dataset: PolypGen (All)** | | | | | |
| U-Net (Ronneberger et al., 2015) | 0.5347 | 0.5995 | 0.6829 | 0.7523 | 0.6105 |
| U-Net++ (Zhou et al., 2018) | 0.5310 | 0.5964 | 0.6765 | 0.7546 | 0.6089 |
| ResU-Net++ (Jha et al., 2019) | 0.3149 | 0.3982 | 0.5887 | 0.4444 | 0.4314 |
| HarDNet-MSEG (Huang et al., 2021) | 0.5376 | 0.6089 | 0.7116 | 0.7124 | 0.6246 |
| ColonSegNet (Jha et al., 2021) | 0.4718 | 0.5486 | 0.6554 | 0.6687 | 0.5617 |
| UACANet (Kim et al., 2021) | 0.5777 | 0.6531 | **0.7493** | 0.7531 | 0.6678 |
| UNeXt (Valanarasu and Patel, 2022) | 0.3761 | 0.4552 | 0.6135 | 0.5600 | 0.4805 |
| **TransNetR (Ours)** | **0.6058** | **0.6668** | 0.7183 | **0.8409** | **0.6706** |
| **Training dataset: Kvasir-SEG − Test dataset: PolypGen (C1)** | | | | | |
| U-Net (Ronneberger et al., 2015) | 0.5772 | 0.6469 | 0.6780 | 0.8464 | 0.6484 |
| U-Net++ (Zhou et al., 2018) | 0.5857 | 0.6611 | 0.6953 | 0.8247 | 0.6700 |
| ResU-Net++ (Jha et al., 2019) | 0.4204 | 0.5239 | 0.6390 | 0.5789 | 0.5557 |
| HarDNet-MSEG (Huang et al., 2021) | 0.6256 | 0.7121 | **0.7800** | 0.7933 | **0.7344** |
| ColonSegNet (Jha et al., 2021) | 0.5514 | 0.6386 | 0.7130 | 0.7423 | 0.6551 |
| UACANet (Kim et al., 2021) | 0.6386 | 0.7189 | 0.7553 | 0.8476 | 0.7254 |
| UNeXt (Valanarasu and Patel, 2022) | 0.4481 | 0.5386 | 0.6421 | 0.6912 | 0.5686 |
| **TransNetR (Ours)** | **0.6538** | **0.7204** | 0.7438 | **0.8778** | 0.7269 |
| **Training dataset: Kvasir-SEG − Test dataset: PolypGen (C2)** | | | | | |
| U-Net (Ronneberger et al., 2015) | 0.5702 | 0.6338 | 0.7347 | 0.7368 | 0.6495 |
| U-Net++ (Zhou et al., 2018) | 0.5612 | 0.6240 | 0.7189 | 0.7631 | 0.6383 |
| ResU-Net++ (Jha et al., 2019) | 0.2779 | 0.3431 | 0.5003 | 0.4198 | 0.3606 |
| HarDNet-MSEG (Huang et al., 2021) | 0.5667 | 0.6311 | 0.7267 | 0.7149 | 0.6376 |
| ColonSegNet (Jha et al., 2021) | 0.4659 | 0.5371 | 0.6443 | 0.6789 | 0.5439 |
| UACANet (Kim et al., 2021) | 0.6091 | 0.6887 | **0.8540** | 0.6870 | 0.7222 |
| UNeXt (Valanarasu and Patel, 2022) | 0.3780 | 0.4583 | 0.6373 | 0.5239 | 0.4837 |
| **TransNetR (Ours)** | **0.6608** | **0.7232** | 0.8071 | **0.8096** | **0.7366** |
| **Training dataset: Kvasir-SEG − Test dataset: PolypGen (C3)** | | | | | |
| U-Net (Ronneberger et al., 2015) | 0.6769 | 0.7481 | 0.7637 | 0.8787 | 0.7518 |
| U-Net++ (Zhou et al., 2018) | 0.6530 | 0.7254 | 0.7526 | 0.8568 | 0.7332 |
| ResU-Net++ (Jha et al., 2019) | 0.4096 | 0.5109 | 0.6463 | 0.5484 | 0.5545 |
| HarDNet-MSEG (Huang et al., 2021) | 0.6623 | 0.7440 | 0.7947 | 0.8180 | 0.7619 |
| ColonSegNet (Jha et al., 2021) | 0.6181 | 0.7064 | 0.7520 | 0.7907 | 0.7221 |
| UACANet (Kim et al., 2021) | 0.7074 | 0.7870 | **0.7954** | 0.8893 | **0.7877** |
| UNeXt (Valanarasu and Patel, 2022) | 0.4654 | 0.5534 | 0.6265 | 0.6868 | 0.5740 |
| **TransNetR (Ours)** | **0.7217** | **0.7874** | 0.7904 | **0.9133** | 0.7863 |

It also helps to better analyze the models' outcomes and investigate for any biased results for specific center data. As observed from Table 5 and Table 6, all models reported similar relative differences on each center as that on the overall PolypGen dataset. Although in some cases, UACANet performed better in recall, TransNetR performed consistently superior and outperformed all the models of each center in mIoU, mDSC and precision. From the qualitative results as well, we can observe that TransNetR is better at segmentation of small, diminutive, sessile, flat and regular polyps (see Figure 4, and Appendix Figure 6).

**Ablation study:** We have presented an ablation study in Table 4 to evaluate the impact of Residual Transformer (RT) block on the TransNetR model. It can be observed that RT block boosts the performance in several performance metrics, such as 1.34% in mIoU, 0.77% in mDSC, 0.02% in recall and 1.5% in precision. As polyp segmentation

is a competitive domain, even smaller improvements in the model performance can make a significant difference in clinical settings by making better disease diagnoses in terms of accuracy and efficiency.

## 4. Conclusion

In this work, we proposed the TransNetR, a transformer-based architecture that utilizes transformer block and residual block to segment polyps with high speed accurately. The experimental results on in-distribution and out-of-distribution data demonstrate the real-time performance of our model with promising polyp segmentation outcomes. A comprehensive comparison of our TransNetR algorithm on 8 centers (6 centers from PolypGen, BKAI-IGH, and CVC-ClinicDB) datasets shows that it consistently outperforms its competitors. The qualitative and quantitative results suggest that TransNetR is more generalizable to the out-of-distribution datasets. The generalization capability of the model makes it suitable for clinical settings, and hence, TransNetR might be a strong benchmark for the development of algorithms that might assist clinicians in early polyp detection.

ACKNOWLEDGEMENT

D. Jha and U. Bagci are supported by the NIH funding: R01-CA246704 and R01-CA240639. V. Sharma is supported by the INSPIRE fellowship (IF190362), Department of Science and Technology, Govt. of India.

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

Table 6: Training dataset: Kvasir-SEG – Test dataset: PolypGen (C4, C5, C6).

| Method | mIoU | mDSC | Recall | Precision | F2 |
|---|---|---|---|---|---|
| **Training dataset: Kvasir-SEG – Test dataset: PolypGen (C4)** | | | | | |
| U-Net (Ronneberger et al., 2015) | 0.3699 | 0.4147 | 0.6550 | 0.5982 | 0.4263 |
| U-Net++ (Zhou et al., 2018) | 0.3807 | 0.4202 | 0.6337 | 0.6099 | 0.4294 |
| ResU-Net++ (Jha et al., 2019) | 0.1689 | 0.2268 | 0.6342 | 0.2816 | 0.2433 |
| HarDNet-MSEG (Huang et al., 2021) | 0.3516 | 0.3936 | 0.6758 | 0.5535 | 0.4062 |
| ColonSegNet (Jha et al., 2021) | 0.2933 | 0.3422 | 0.6493 | 0.4710 | 0.3558 |
| UACANet (Kim et al., 2021) | 0.4273 | 0.4828 | **0.7371** | 0.6301 | 0.4982 |
| UNeXt (Valanarasu and Patel, 2022) | 0.2261 | 0.2757 | 0.6645 | 0.3552 | 0.2989 |
| **TransNetR (Ours)** | **0.4601** | **0.5042** | 0.6874 | **0.7141** | **0.5096** |
| **Training dataset: Kvasir-SEG – Test dataset: PolypGen (C5)** | | | | | |
| U-Net (Ronneberger et al., 2015) | 0.2963 | 0.3614 | 0.4577 | 0.5497 | 0.3870 |
| U-Net++ (Zhou et al., 2018) | 0.3143 | 0.3773 | 0.4475 | 0.6030 | 0.3935 |
| ResU-Net++ (Jha et al., 2019) | 0.2041 | 0.2748 | 0.4643 | 0.3027 | 0.3156 |
| HarDNet-MSEG (Huang et al., 2021) | 0.3090 | 0.3769 | 0.4588 | 0.5250 | 0.3970 |
| ColonSegNet (Jha et al., 2021) | 0.2687 | 0.3416 | 0.4097 | 0.5232 | 0.3532 |
| UACANet (Kim et al., 2021) | 0.3257 | 0.4028 | **0.4941** | 0.5615 | **0.4250** |
| UNeXt (Valanarasu and Patel, 2022) | 0.2530 | 0.3288 | 0.4646 | 0.4192 | 0.3583 |
| **TransNetR (Ours)** | **0.3597** | **0.4214** | 0.4508 | **0.7767** | 0.4232 |
| **Training dataset: Kvasir-SEG – Test dataset: PolypGen (C6)** | | | | | |
| U-Net (Ronneberger et al., 2015) | 0.5384 | 0.6126 | 0.7054 | 0.7508 | 0.6362 |
| U-Net++ (Zhou et al., 2018) | 0.5355 | 0.6163 | 0.7340 | 0.7230 | 0.6564 |
| ResU-Net++ (Jha et al., 2019) | 0.2816 | 0.3684 | 0.6220 | 0.3526 | 0.4326 |
| HarDNet-MSEG (Huang et al., 2021) | 0.5548 | 0.6341 | 0.7197 | 0.7722 | 0.6487 |
| ColonSegNet (Jha et al., 2021) | 0.4410 | 0.5290 | 0.6199 | 0.6403 | 0.5424 |
| UACANet (Kim et al., 2021) | 0.6039 | 0.6748 | **0.7698** | 0.7669 | **0.7028** |
| UNeXt (Valanarasu and Patel, 2022) | 0.3743 | 0.4539 | 0.6019 | 0.5045 | 0.4850 |
| **TransNetR (Ours)** | **0.6335** | **0.6917** | 0.6783 | **0.9431** | 0.6803 |

## Appendix A. Additional Results

Figure 6 (a) shows the example of the input image from out-of-distribution (unique medical centers (center 6 from PolypGen)), corresponding ground truth, predicted masks, and the heatmap of the intermediate feature maps of the TransNetR. The prediction results show that TransNetR is better at predicting different-sized polyps.

Figure 6 (b) shows the example of the input image from out-of-distribution (unique medical centers), corresponding ground truth, predicted masks, and the heatmap of the intermediate feature maps of the TransNetR. In the provided heatmap, the "red" and "yellow" areas represent essential features of TransNetR, and the "blue" area refers to the features that are not significantly important. The prediction shows that TransNetR can perform well even on small polyps, medium or regular polyps, and even on image frames with more than one polyp.

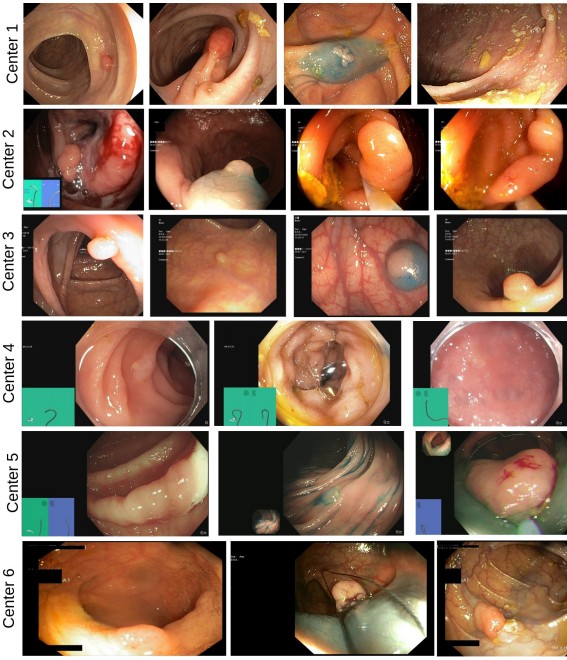

Figure 5: Center-wise example images from the PolypGen dataset. Here, the variability among the dataset from different centers can be observed. There is a difference in image resolutions and sizes, shapes, colors, textures and appearances and collection protocols.

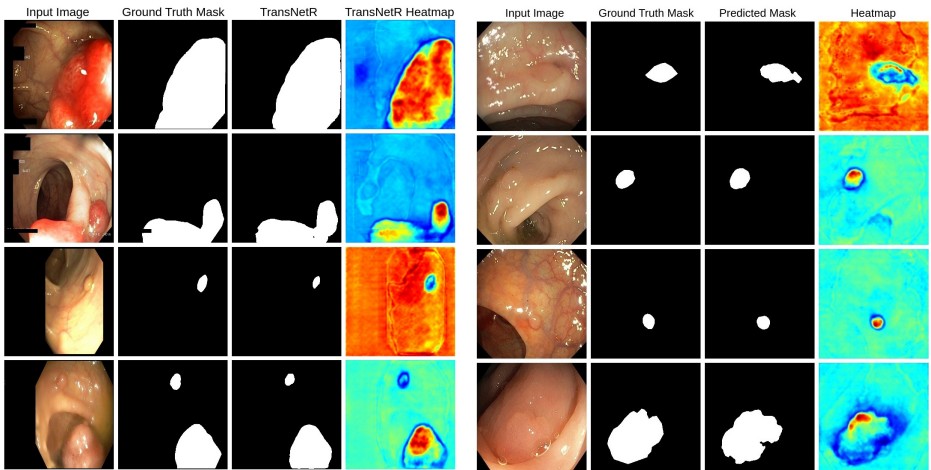

Figure 6: Qualitative result when the TransNetR is trained on Kvasir-SEG and tested on (a) PolypGen (center 6 (C6)) and (b) PolypGen (center 1 (C1)).

