# OpenReview forum: "TransNetR: Transformer-based Residual Network for Polyp Segmentation with Multi-Center Out-of-Distribution Testing"
_MIDL.io/2023/Conference — MIDL 2023 Poster_

### Official Review · Reviewer_78gs · 2023-01-31

**Confidence:** 4
**Preliminary Rating:** 3

**Summary:**

The authors propose to utilize pre-trained ResNet50 as the encoder and insert transformer blocks in the decoder to build Transformer based Residual network (TransNetR) for colon polyp segmentation. The authors evaluate their method on the Kvasir-SEG dataset to show their method can achieve state-of-art results. The authors also evaluate their method on the out-of-distribution dataset, where the distribution is unknown and different from the training distribution, to show the generalizability of their TransNetR.

**Strengths:**

(1) The idea of utilizing transformer blocks (attention blocks) for colon polyp segmentation seems to be novel.

(2) The authors evaluate their method not only with in-distribution test data but also with unseen out-of-distribution test data.

(3) The proposed method can achieve real-time processing speed, which is crucial for clinical integration.

**Weaknesses:**

(1) The major weakness of this paper is the lack of comparison with other transformer or attention-based methods. Especially when the proposed method is kind of similar to previous work. Please refer to my detailed comments on this.

(2) The claim of previous work lacking generalizability in the Introduction section seems incorrect. Please refer to my detailed comments on this.

(3) No ablation studies are conducted on the design of the architecture.

**Deanonymize Review:**

no

**Detailed Comments:**

For weakness point (1): The idea of utilizing transformer blocks or attention blocks in the design to improve performance at first glance is novel. However, there exists many previous works that integrate transformer blocks or attention blocks in their methods. Without any comparisons, it is difficult to judge the performance of the proposed methods. Below I shared many papers that focus on utilizing transformer blocks or attention blocks to improve performance for image segmentation.  Some authors utilize the transformer as the encoder, some authors put attention blocks in the decoder, some authors insert the transformer in the middle of the encoder and decoder, and some authors put transformer blocks in the connection between the encoder and decoder.

[1] Sanderson, Edward, and Bogdan J. Matuszewski. "FCN-transformer feature fusion for polyp segmentation." Medical Image Understanding and Analysis: 26th Annual Conference, MIUA 2022, Cambridge, UK, July 27–29, 2022, Proceedings. Cham: Springer International Publishing, 2022.

[2] Mandujano-Cornejo, Vittorino, and Javier A. Montoya-Zegarra. "Polyp2Seg: Improved Polyp Segmentation with Vision Transformer." Medical Image Understanding and Analysis: 26th Annual Conference, MIUA 2022, Cambridge, UK, July 27–29, 2022, Proceedings. Cham: Springer International Publishing, 2022.

[3] Duc, Nguyen Thanh, et al. "ColonFormer: an efficient transformer based method for colon polyp segmentation." IEEE Access 10 (2022): 80575-80586.

[4] Patel, Krushi Bharatbhai, Fengjun Li, and Guanghui Wang. "FuzzyNet: A Fuzzy Attention Module for Polyp Segmentation." NeurIPS'22 Workshop on All Things Attention: Bridging Different Perspectives on Attention.

[5] Li, Bo, et al. "RT‐Unet: An advanced network based on residual network and transformer for medical image segmentation." International Journal of Intelligent Systems 37.11 (2022): 8565-8582.

[6] Wang, Haonan, et al. "Uctransnet: rethinking the skip connections in u-net from a channel-wise perspective with transformer." Proceedings of the AAAI conference on artificial intelligence. Vol. 36. No. 3. 2022.

[7] Chen, Jieneng, et al. "Transunet: Transformers make strong encoders for medical image segmentation." arXiv preprint arXiv:2102.04306 (2021).


For weakness point (2): The authors claim previous work [3] lacks generalizability in paragraph 3 of the Introduction section. In Table 3 of [3], ColonFormer trained with the CVC-ClinicDB dataset and performed well when tested with the ETIS-Larib dataset.


For weakness point (3): The authors did not share any ablation studies with the design of TransNetR. Below are some examples of the ablation studies for TransNetR.

(a) Parameter settings for the transformer encoder block in Residual Transformer Block (RT Block)

(b) How many RT Block should be utilized (Maybe consider including no RT block is utilized, which is no Transformer is utilized)

(c) Where insert this RT Block (the decoder, in the middle between encoder and decoder, or the connection between encoder and decoder)

**Paper Type:**

both

**Questions To Address In The Rebuttal:**

For weakness point (1), maybe consider comparing the proposed method with other transformer or attention-based methods. Note that residual axial attention block is proposed in [3].

For weakness point (2), maybe compare with [3] to justify the statement.

For weakness point (3), maybe consider adding ablation studies to justify the design choices. This weakness point is less important than the weakness point (1).

---

### Official Review · Reviewer_FMum · 2023-02-02

**Confidence:** 4
**Preliminary Rating:** 3
**Recommendation:** Poster

**Summary:**

The paper introduces TransNetR model which consists of pretrained resnet and decoder with residual transformer blocks. The method achieves realtime inference speed and is evaluated using 4 datasets. The method showed satisfactory generalization ability across study centers. The source code is also available.

**Strengths:**

- the method has good generalization ability, multicenter testing
- the source code is publicly available on github
- the method achieves state of the art performance and is capable of realtime inference


**Weaknesses:**

- it's not clear why the three testing datasets are OOD. Do they present different types of polyps? It would be helpful to know what exactly is different in the testing data. if it's lighting conditions, camera, etc.. I wouldn't call it OOD but a generalization study.
- In addition to FPS it would be interesting to know parameters count and FLOPs of the models.
- page 8 section Generalization ability: the phrase "noticeable performance drop on the OOD datasets" requires quantitative support.

**Deanonymize Review:**

no

**Paper Type:**

methodological development

**Questions To Address In The Rebuttal:**

In the rebuttal, I would like to see what makes the test datasets to be out-of-the distribution. As well as additional quantitative information such as parameters count, FLOPS, and "performance drop on the OOD datasets" would be appreciated.

---

### Official Review · Reviewer_7sEk · 2023-02-05

**Confidence:** 5
**Preliminary Rating:** 3

**Summary:**

This paper develops a novel real-time Transformer based Residual network for colon polyp segmentation and evaluate its diagnostic performance. Apart from this, it explores the generalizability of the network by testing on the OoD dataset. The motivation of this article is clear, and the performance has been improved to a certain extent, but its innovation points are not prominent, and the analysis of why it can achieve real-time segmentation and strong generalization ability for OOD data is insufficient.

**Strengths:**

The TransNetR integrates the strength of transformer and residual learning to generate precise segmentation outcomes even while testing on OOD data and maintains high performance along with real-time processing speed.


**Weaknesses:**

1) The author repeatedly mentioned the generalization ability and real-time performance of the network structure, and the structural design of the network for this needs to be further strengthened.
2) The design of Transformer-based Residual is not very novel yet. There are currently many types of residual design based on transformers [1-3]. What are your special features, similarities and differences?
[1] ResViT: residual vision transformers for multimodal medical image synthesis
[2] RTUNet: Residual transformer UNet specifically for pancreas segmentation
[3] RT‐Unet: An advanced network based on residual network and transformer for medical image segmentation.
3) More SOTA methods are not listed in the Tab. 3, such as Polyp-PVT, UACANet and PraNet.

**Deanonymize Review:**

no

**Detailed Comments:**

1) It seems that most of the real-time and generalization capabilities come from the contribution of the basic network. How to determine the contribution of the Transformer-based Residual Network? I didn't see a special design for lightweight or generalization.
2) Experiments and analysis on real-time and generalization still need to be strengthened.
3) Can TransNetR be migrated to Video Polyp segmentation? What about its real-time performance?

**Paper Type:**

methodological development

**Questions To Address In The Rebuttal:**

1) The author repeatedly mentioned the generalization ability and real-time performance of the network structure, and the structural design of the network for this needs to be further strengthened.
2) The design of Transformer-based Residual is not very novel yet. There are currently many types of residual design based on transformers [1-3]. What are your special features, similarities and differences?
[1] ResViT: residual vision transformers for multimodal medical image synthesis
[2] RTUNet: Residual transformer UNet specifically for pancreas segmentation
[3] RT‐Unet: An advanced network based on residual network and transformer for medical image segmentation.
3) More SOTA methods are not listed in the Tab. 3, such as Polyp-PVT, UACANet and PraNet.
4) It seems that most of the real-time and generalization capabilities come from the contribution of the basic network. How to determine the contribution of the Transformer-based Residual Network? I didn't see a special design for lightweight or generalization.
5) Experiments and analysis on real-time and generalization still need to be strengthened.
6) Can TransNetR be migrated to Video Polyp segmentation? What about its real-time performance?

---

### Meta-Review · Area_Chair_Rkqe · 2023-02-24

**Recommendation:** Accept (Poster)
**Confidence:** 5

**Metareview:**

The authors have partially addressed the reviewers' concerns, i.e. motivation, experimental performance. Overall, I satisfy the proposed paper. How to deal with multi-center OOD during model deploying is a useful topic.
One review mentioned it should be compared with other SOTA baselines. Due to the time limitation of rebuttal, I think the authors have tried their best to reply the comment. Moreover, for rebuttal, reviewers should refrain from requesting significant additional experiments for the rebuttal, or penalize for lack of additional experiments.